# Empagliflozin Reduces Interleukin-6 Levels in Patients with Heart Failure

**DOI:** 10.3390/jcm12134458

**Published:** 2023-07-03

**Authors:** Michael Gotzmann, Pauline Henk, Ulrik Stervbo, Arturo Blázquez-Navarro, Andreas Mügge, Nina Babel, Timm H. Westhoff

**Affiliations:** 1University Hospital St Josef-Hospital, Cardiology and Rhythmology, Ruhr University Bochum, 44791 Bochum, Germany; 2University Hospital Marien Hospital Herne, Medical Department 1, Ruhr University Bochum, 44625 Herne, Germanytimm.westhoff@elisabethgruppe.de (T.H.W.)

**Keywords:** SGLT-2-inhibitors, empagliflozin, heart failure, interleukin 6

## Abstract

**Background:** The inhibition of sodium-glucose co-transporter 2 (SGLT-2) has been shown to be beneficial in the treatment of diabetic and non-diabetic patients with heart failure. The underlying mechanisms are incompletely understood. The present prospective study investigates for the first time the effect of empagliflozin on various soluble markers of inflammation in patients with reduced ejection fraction (HFrEF). **Methods:** We included 50 inpatients with HFrEF and diabetes mellitus type 2. A total of 25 patients received a therapy with the SGLT-2-inhibitor empagliflozin in addition to standard medication; the other 25 patients did not receive empagliflozin and were considered the control group. Quality of life, functional status and soluble immunological parameters in serum were assessed at baseline and after 3 months. **Results:** The baseline characteristics of both groups revealed no significant differences. Patients on empagliflozin demonstrated a significant improvement in the Minnesota living with heart failure questionnaire (baseline 44.2 ± 20.2 vs. 24 ± 17.7; *p* < 0.001), in distance in the 6-min walk test (baseline 343 ± 145 m vs. 450 ± 115 m; *p* < 0.001) and in soluble interleukin-6 level (baseline 21.7 ± 21.8 pg/mL vs. 13.7 ± 15.8 pg/mL; *p* = 0.008). There was no significant change of these or other parameters in the control group (*p* > 0.05 each). **Conclusions:** The empagliflozin-induced improvement of quality of life and functional capacity in patients with HFrEF and type 2 diabetes mellitus is accompanied by a substantial reduction of interleukin-6 levels. Thus, anti-inflammatory effects may contribute to the benefits of SGLT-2-inhibitors in heart failure.

## 1. Introduction

Heart failure is a frequent long-term complication of diabetes mellitus type 2. Coexisting diabetes mellitus significantly deteriorates the prognosis of patients with heart failure [1]. Diabetes mellitus can directly damage the myocardium by causing macro- and microangiopathy of the coronary vessels. In addition to vasculopathy, several pathophysiological mechanisms of diabetic cardiomyopathy are discussed, including the formation of advanced glycation end-products, increased oxidative stress, increased neurohumoral activation and, in particular, inflammation [2]. Inflammation plays a central role in the onset and progression of heart failure [3].

Randomized trials demonstrated the beneficial effects of sodium-glucose co-transporter 2 (SGLT-2) inhibitors in patients with type 2 diabetes mellitus and heart failure with reduced ejection fraction [4,5,6]. These favourable effects have been also demonstrated in non-diabetic patients with heart failure and reduced ejection fraction (HFrEF) [7,8,9] and in patients with heart failure with preserved ejection fraction (HFpEF) [9,10].

The beneficial effects of SGLT-2-inhibitors on heart failure are still under discussion. SGLT-2-inhibitors act as diuretics in the proximal tubule and thereby reduce states of hypervolemia. Moreover, they modify intrarenal hemodynamics, induce a slight decrease of blood pressure and induce a metabolic shift to mild ketonemia. The majority of these effects may be elicited by other diuretics as well, but without yielding the outstanding benefits observed in SGLT-2-inhibitors trials. Thus, it must be hypothesized that there are undetected additional mechanisms, e.g., anti-inflammatory effects [11,12,13].

Previous studies examining liver and kidney function demonstrated that SGLT-2-inhibitors can reduce organ inflammation [3]. Although it is not clear which myocardial cells are most involved in inflammation in HF, it has been demonstrated in cultured myocytes that SGLT-2-inhibitors are potent in reducing the transcript levels of numerous markers of inflammation [14]. Moreover, studies in recent years suggest that SGLT-2-inhibitors are suitable for reducing oxidative stress and endothelial dysfunction as well as improving cardiac metabolism and energetics [3].

However, data on the immunological and inflammatory responses to SGLT-2-inhibitors in patients with HFrEF are sparse. Therefore, the aim of this prospective controlled proof-of-concept study is to investigate whether the SGLT-2-inhibitor empagliflozin elicits an immunomodulatory effect in patients with HFrEF. For this aim, we assessed a broad profile of soluble immunological mediators before and three months after initiation of SGLT-2-inhibitors and compared the results to a control group without an SGLT-2-inhibitor.

## 2. Methods

### 2.1. Study Population

From June 2020 to September 2021, we enrolled hospitalized patients treated for heart failure in the Department of Cardiology/Rhythmology at St. Josef Hospital (Hospital of the Ruhr-University Bochum, Bochum, Germany). Thus, when the study was designed, SGLT-2-inhibitors were approved for diabetes mellitus type 2 with heart failure with reduced, but not preserved, ejection fraction.

Inclusion criteria were: symptomatic left heart failure (NYHA stage II-IV), left ventricular ejection fraction ≤ 40%, type 2 diabetes mellitus, life expectancy > 1 year, informed consent and age ≥ 18 years. Exclusion criteria were: any immunosuppressive therapy, fever and clinical signs of an ongoing infectious disease, ongoing or former therapy with an SGLT-2-inhibitor, type 1 diabetes mellitus, renal insufficiency (GFR < 45 mL/min/1.73 m^2^), hypotension or systolic blood pressure < 90 mmHg, myocardial infarction in the 30 days prior to study inclusion, known peripheral arterial disease and insufficient compliance that makes regular use of medication unlikely.

Patients were clinically in a stable condition at the time of study inclusion and were willing to participate in this prospective non-interventional proof-of-concept study. Written informed consent was obtained from all participants. The study was approved by the local ethics committee of the Ruhr University Bochum (reg. number 20-6944).

### 2.2. Study Protocol

We performed a prospective, controlled, non-randomized study. Study investigations particularly included quality of life, functional status, left ventricular ejection fraction and immunological factors.

The Minnesota Living with Heart Failure Questionnaire (MLHFQ) was designed to measure the impact of heart failure and heart failure treatment on the physical, emotional, social and mental dimensions of quality of life. The sum of the responses to each item yields the total MLHFQ score for each patient. The test score ranges from 0 to 110, with a higher score indicating poorer quality of life [15]. An adaptation of the MLHFQ for German-speaking patients was used.

The 6-min walk test was performed according to American Thoracic Society guidelines for a functional test. Patients were instructed to walk rapidly for 6 min or until the onset of dyspnea or muscle fatigue. The walking distance was 30 m. The total walking distance was recorded [16].

Echocardiographic imaging was performed on the same day as the clinical examinations. Transthoracic echocardiography was obtained according to the guidelines of the American Society of Echocardiography and European Association of Cardiovascular Imaging [17] using a digital ultrasound machine (Vivid 9, General Electrics, Horton, Norway). Left ventricular ejection fraction was measured using the Simpson method in the 4- and 2-chamber views. The echocardiographic examination was performed by a single observer.

Blood samples were taken on the day of the study investigations. The estimated glomerular filtration rate was calculated using the abbreviated Modification of Diet in Renal Disease Study equation. Kidney disease was defined as a glomerular filtration rate of 60 mL/min/1.73 m^2^.

Soluble factors were analysed as previously described by Stervbo et al. [18]. Briefly, soluble mediators including human IFN-α2, IFN-γ, TNF-α, MCP-1, IL-6, IL-8, IL-10, IL-12p70, IL-17A, IL-18, and IL-23 were assessed using the LEGENDplex custom panel (BioLegend, San Diego, CA, USA). The samples were processed following the manufacturer’s instructions. The analyte concentration was extracted using LEGENDplex Data Analysis Software. The analysis of the immunological factors was blinded to the clinical parameters.

### 2.3. SGLT-2-inhibitors and Follow-Up

Prescription of empagliflozin was done at the advice of the attending physicians not involved in this study. Usually, the indication for empagliflozin was optimizing the glucose control. Since all patients in this study had never received SGLT-2-inhibitors previously, the recommended initial dose of 10 mg empagliflozin per day was prescribed for all patients. The results of the examinations for quality of life, functional status, and immunological factors were not known to the treating physicians.

The examinations (MLHFQ, 6-min walk test, laboratory tests of immunological parameters, and echocardiographic determination of left ventricular ejection fraction) were repeated at an outpatient visit after 3 months.

## 3. Statistics

Results are presented as mean ± standard deviation (SD) or standard error of mean where appropriate. Continuous variables were compared between groups (with and without SGLT-2-inhibitors) using an unpaired t test (for normally distributed variables) or Mann–Whitney U test (for non-normally distributed variables). χ2 analysis was used to compare categoric variables.

The continuous variables were compared at baseline and after 3 months using the paired Student t test for normally distributed variables or the Wilcoxon test for non-normally distributed variables. Post hoc multiple comparison tests were performed using the Benjamini–Hochberg adjustment. A *p* value < 0.05 was considered significant. All probability values reported are 2-sided.

## 4. Results

In this study, we enrolled a total of 50 hospitalised patients (11 women, 39 men) with HFrEF and diabetes mellitus type II. The mean age of the patients was 71 ± 11 years, the mean left ventricular ejection fraction was 30.7 ± 8.7% and the mean HbA1c was 7.8 ± 1.8%. Coronary artery disease was present in 30 patients (Table 1).

The six-minute walk test could only be performed in 43 of the 50 study patients due to orthopaedic problems or walking disabilities. All other examinations were performed in all patients at baseline and 3 months later.

After enrolment, 25 patients received the SGLT-2-inhibitor empagliflozin (dose 10 mg daily) in addition to their previous therapy at the decision of the treating physician. In the remaining 25 patients, considered the control group, no SGLT-2-inhibitor was prescribed.

The clinical characteristics, medication, MLHFQ results, distance of the 6-min walk test and echocardiographic parameters are given in Table 1 and Table 2. The results of the immunological tests are listed in Table 3. Although the present study was not a randomized trial, the patients with and without SGLT-2-inhibitor demonstrated no significant differences at baseline (Table 1, Table 2 and Table 3).

The results of the study-specific tests (MLHFQ, 6-min walk test, left ventricular ejection fraction and immunologic factors) at baseline and after 3 months are presented in Table 4.

Patients receiving empagliflozin presented better quality of life, increased walking distance in the 6-min walk test, improved left ventricular ejection fraction, reduced C-reactive protein and reduced interleukin-6 levels at 3 months compared with baseline (Table 4). In comparison, the control group (patients without empagliflozin) demonstrated only an improvement in quality of life, left ventricular ejection fraction and reduced C-reactive protein levels. Notably, the other immunological factors revealed no difference in these patients (Table 4).

## 5. Discussion

The main finding of this prospective proof-of-concept study is that three months of empagliflozin in patients with HFrEF and diabetes mellitus type II is associated with a reduction in soluble interleukin-6 levels. In contrast, patients without therapy with empagliflozin revealed no change in their soluble immune mediators (Table 4). In addition, the use of an SGLT-2-inhibitor was also associated with an improvement in quality of life, functional capacity and left ventricular ejection fraction (Figure 1). Our study therefore supports the hypothesis that part of the effect of SGLT-2-inhibitors may be due to positive immunological effects.

In recent years, there have been prospective randomized trials demonstrating that anti-inflammatory therapy leads to improved cardiovascular outcomes in patients with coronary artery disease [19,20]. This may indicate a general mechanism and could improve the treatment of cardiovascular disease.

A number of studies have demonstrated the beneficial effects of SGLT-2-inhibitors on diabetes mellitus but also on renal function and cardiovascular disease [4,5,6,7,8,9,10]. The exact mechanism of these relatively new agents remains elusive. Various hemodynamic, metabolic and immunologic effects have been described in the past [11,12,13].

Rodent models suggest that SGLT2 is only expressed in the proximal tubular cells of the kidney [21]. Hence, it has remained elusive whether SGLT-2-inhibitors can indeed have anti-inflammatory effects. If so, they are not supposed to be mediated by direct effects on immune cells but rather secondary effects following improved hemodynamics, mild ketonemia or reduction of water overload. Our findings are in line with recent findings of the CANVAS study program. In analogy to our findings with empagloflozin, canagliflozin reduced IL6 concentrations in patients with type 2 diabetes at high cardiovascular risk [22]. Interestingly, baseline IL-6 and its 1-year change were associated with renal and cardiovascular outcomes. Although this association is not necessarily causal, it shows that IL-6 may be used as a prognostic biomarker in this context.

Moreover, increased ketone body concentrations have been suggested to be responsible for any anti-inflammatory effect [23]. On the other hand, there is evidence for an interaction between cytokines and the glucose transporter SGLT-2 [24,25]. Empagliflozin also attenuated the secretion and mRNA expression of proinflammatory cytokines, such as tumour necrosis factor-α, Interleukin-1β, Interleukin-6 and Interferon-γ, and proinflammatory chemokines [26]. Kim et al. also investigated the effect of SGLT-2-inhibitors on macrophages. The authors concluded that the SGLT-2-inhibitor attenuates pyrin domain-containing 3 inflammasome activation, which may explain its cardioprotective effects [27]. A different approach was taken by Kolijn et al. in their in vitro study in human and murine cardiac myocytes with heart failure with preserved ejection fraction. In this study, empagliflozin significantly suppressed elevated levels of pro-inflammatory cytokines and attenuated pathological oxidative parameters in both cardiomyocyte cytosol and mitochondria [28]. In a secondary analysis of EMPA-TROPISM [ATRU-4], 6 months of empagliflozin significantly improved body mass index, interstitial myocardial fibrosis, aortic stiffness and inflammatory markers. In proteomic analysis, 92 proteins were studied. The empagliflozin group showed significant changes in the expression of 17 proteins (proteins mostly involved in inflammatory processes) compared with the placebo group [29].

These previous (mainly in vitro) studies provide evidence that immunological parameters are affected by the use of SGLT-2-inhibitors and, in particular, empagliflozin. This effect seems to be consistent in different types of heart failure with preserved [28] and reduced ejection fraction [29] and in diabetic [27] and non-diabetic patients [29]. However, there have been no studies investigating the changes in immunological parameters in vivo in patients with heart failure.

At the beginning of our study, SGLT-2 inhibitors were approved only for the treatment of HFrEF and type 2 diabetes mellitus. Accordingly, we included only patients with these diseases. Consistent with previous studies, our study demonstrated an effect of therapy with empagliflozin on immunological parameters. However, in contrast to previous studies, we were able to demonstrate for the first time this effect in vivo by determining the soluble immune mediators in serum (Table 4).

As a limitation, it should be noted, that of the 14 factors analysed, only C-reactive protein and soluble Interleukin-6 showed a significant change. Interestingly, the level of C-reactive protein decreased in the groups of patients both with and without empagliflozin. However, the change in C-reactive protein was no longer significant in both groups after adjustment for multiple testing (Table 4). The fact that other immunological parameters did not exhibit a change could be due to the relatively small number of patients and the relatively short duration of therapy with empagliflozin. It is therefore conceivable that with a larger study cohort and a longer study period, an additional significant change in further immunological parameters might be detectable. Nevertheless, our study confirms observations made in previous studies. A reduction in mRNA expression of Interleukin-6 in empagliflozin-treated macrophages [26] and intracellular levels in human and murine myocardium [28] have already been described.

Interleukin-6 has a key role in innate immunity. The action of this cytokine is particularly related to host defence, regulation, proliferation and differentiation of immune cells [30]. Interleukin-6, which is detectable in blood, originates from various sources, mainly mononuclear macrophages, but also T-helper cells, B-cells, vascular endothelial cells, smooth muscle cells and fibroblasts [31].

For several years, it has been known that the expression of interleukin-6 is increased in the circulation and myocardium of patients with heart failure. Similarly, interleukin-6 is known to be associated with the progression of heart failure. Inflammation-induced remodelling of the myocardium (including an increase in apoptosis of myocytes and a decrease in contractility) is thought to be responsible [32,33]. Elevated blood interleukin-6 concentrations have also been associated with increased heart failure–related mortality [34].

As pointed out, previous experimental analyses suggest that SGLT-2-inhibitors have an immunomodulatory effect [26,27,28,29]. Nevertheless, it remains conceivable that the reduction in soluble interleukin-6 that we observed is mediated by other beneficial effects of empaglifozin. Of importance in this context is the above-mentioned observation of Ghezzi et al., whose study mapped the distribution of functional SGLT-2 proteins in rodents using positron emission tomography with 4-[18F]fluorodapagliflozin (F-Dapa). Microscopic ex vitro autoradiography of the kidney revealed binding of F-Dapa to the proximal tubules. Of note, in vivo imaging demonstrated no measurable specific binding of F-Dapa in the heart, muscle, salivary glands, liver or brain. The high renal specificity of SGLT-2-inhibitors may indicate that there is primarily a renal mechanism of these drugs [21].

In our study population with 3 months of therapy with empagliflozin, there was a clear, clinically relevant difference in quality of life and distance in the 6-min walk test. This clinical benefit was associated with a decrease of Interleukin-6 concentrations. These findings are of practical clinical interest, since they show that SGLT-2 inhibition indeed has anti-inflammatory properties, although the transporter has not yet been identified in immune cells. They thereby contribute another piece in the puzzle in our understanding of the complex effects of SGLT-2-inhibition. Second, in line with the recent insights from CANVAS, the study identifies Interleukin-6 as a potential biomarker in the evaluation of therapeutic measures in heart failure [22]. Future SGLT-2-inhibitor studies should characterize this broadly available biomarker with regard to prognosis and therapy guidance in heart failure. Moreover, it will be of interest whether SGLT-2-inhibitors reduce Interleukin-6 in non-heart-failure populations as well, e.g., in chronic kidney disease.

## 6. Limitations

In this study, only the SGLT-2-inhibitor empagliflozin was used. It is therefore not clear whether other SGLT-2-inhibitors have a similar effect. Another limitation is the fact that this is not a randomized, double-blinded study. Even if there was no randomisation, however, the control group did not show any changes in quality of life, physical capacity or inflammatory parameters, therefore making a false-positive result unlikely. Moreover, the analysis of the immunological factors was performed without knowledge of the clinical parameters.

The main limitation of the study is the small number of patients. However, this study is a proof-of-concept study to determine whether the use of SGLT-2-inhibitors leads to a significant effect of different soluble immune mediators. Further studies should confirm the effects on soluble Interleukin-6 and other immunological factors.

## 7. Conclusions

Three months of empagliflozin significantly improved the quality of life and functional capacity in patients with heart failure with reduced ejection fraction and type 2 diabetes mellitus. Simultaneously, there was a reduction in soluble Interleukin-6 levels, an effect that was not demonstrated in a corresponding control group. Hence, our study provides the first in vivo evidence of the potential anti-inflammatory effect of SGLT-2-inhibition in patients with heart failure. Our study confirms preliminary in vitro data on the immunomodulatory effects of SGLT-2-inhibitors. Whether the reduction of Interleukin-6 levels is a mechanism for clinical improvement in patients or a marker of heart failure severity should be the subject of further research.

## Figures and Tables

**Figure 1 jcm-12-04458-f001:**
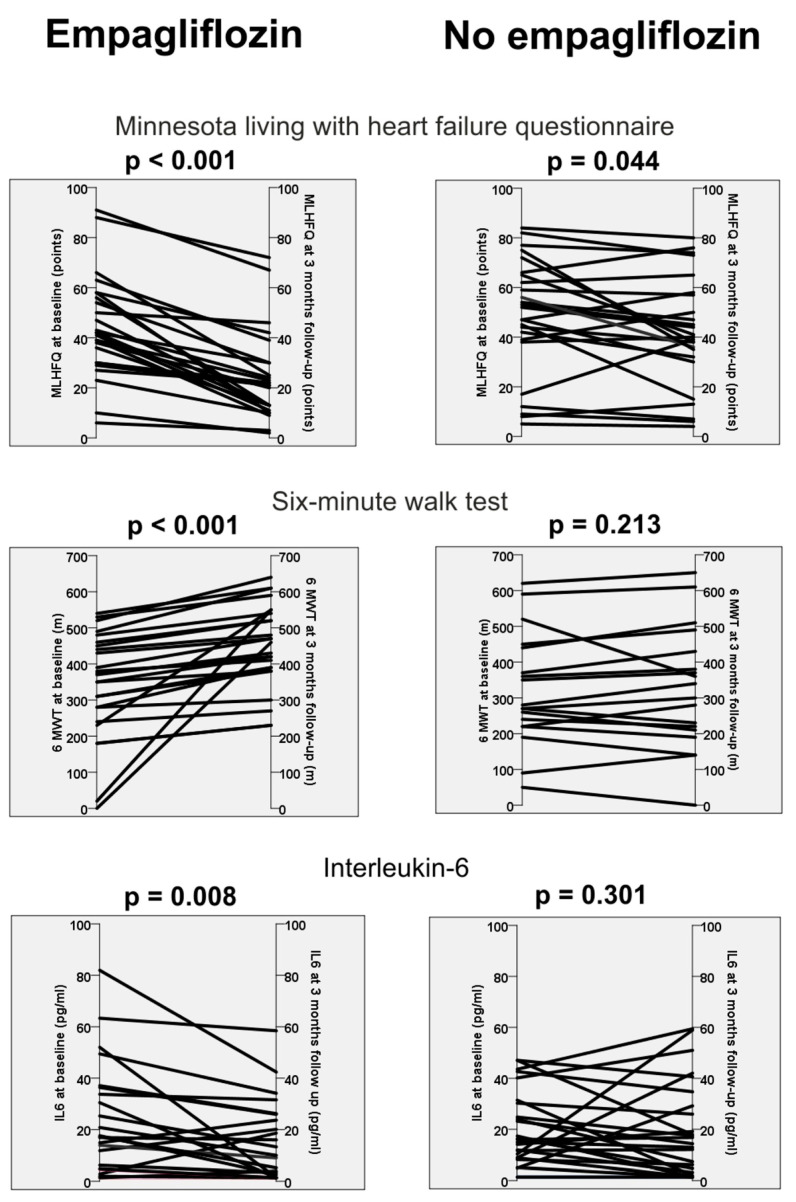
Minnesota living with heart failure questionnaire, six-minute walk test and Interleukin-6 at baseline and at 3 months follow-up in patients with empagliflozin and without empagliflozin.

**Table 1 jcm-12-04458-t001:** Clinical characteristics of study patients (*n* = 50).

	Empagliflozin (*n* = 25)	No Empagliflozin (*n* = 25)	*p* Value
Age (years)	67.9 ± 11.2	74.1 ± 10.4	0.079
Women (♀), *n* (%)	4 (16)	7 (28)	0.306
Body mass index (kg/m^2^)	32.8 ± 7	29.7 ± 6	0.184
NYHA functional class			0.393
NYHA II, *n* (%)	4 (16)	4 (16)	
NYHA III, *n* (%)	17 (68)	13 (52)	
NYHA IV, *n* (%)	4 (16)	8 (32)	
Medical history			
Coronary artery disease, *n* (%)	18 (72)	12 (48)	0.083
Previous hospitalizations due to heart failure, *n* (%)	19 (76)	17 (68)	0.682
ICD, *n* (%)	9 (36)	10 (40)	0.771
CRT, *n* (%)	3 (12)	5 (20)	0.440
Atrial fibrillation, *n* (%)	13 (52)	15 (60)	0.569
Peripheral artery disease, *n* (%)	3 (12)	2 (8)	0.637
Previous stroke/TIA, *n* (%)	3 (12)	5 (20)	0.440
Medication			
	23 (92)	21 (84)	0.384
ACE-Inhibitors and ARB, *n* (%)	15 (60)	18 (72)	0.370
Sacubitril/valsartan, *n* (%)	8 (33)	5 (20)	0.291
Aldosterone antagonists, *n* (%)	9 (36)	7 (28)	0.544
Loop diuretics, *n* (%)	15 (60)	19 (76)	0.225
Insulin therapy, *n* (%)	8 (32)	4 (16)	0.185
Metformin, *n* (%)	14 (56)	9 (36)	0.156

NYHA, New York Heart Association; ICD, implantable cardioverter-defibrillator; CRT, cardiac resynchronization therapy; TIA, transient ischemic attack; ACE, angiotensin-converting enzyme; ARB, angiotensin II receptor blockers.

**Table 2 jcm-12-04458-t002:** Physical examination, ECG, laboratory parameters and echocardiography at baseline.

	Empagliflozin (*n* = 25)	No Empagliflozin (*n* = 25)	*p* Value
Systolic blood pressure (mmHg)	125 ± 14	122 ± 19	0.432
Diastolic blood pressure (mmHg)	80 ± 12.1	78 ± 10.4	0.586
MLHFQ (points)	44.2 ± 20.2	48.4 ± 23.4	0.264
Six-minute walk test (m) *	343 ± 145	321 ± 157	0.395
ECG			
Heart rate (bpm)	85 ± 23	76 ± 25	0.226
QRS (ms)	120 ± 26	126 ± 36	0.708
Left bundle branch block, *n* (%)	6 (24)	6 (24)	1.000
Laboratory			
HbA1c (%)	8.0 ± 1.8	7.6 ± 1.9	0.206
Hemoglobin (g/dL)	13.6 ± 2.1	13.1 ± 1.9	0.461
GFR (mL/min/1.73 m^2^)	68.9 ± 19.5	62 ± 14	0.210
Echocardiographic parameters			
LVEF (%)	32.5 ± 8.4	28.8 ± 8.9	0.177
Left atrial diameter (mm)	43.6 ± 7	44.7 ± 5.1	0.711
Mitral regurgitation			0.481
None/trace, *n* (%)	4 (16)	7 (28)	
Mild, *n* (%)	17 (68)	13 (52)	
Moderate, *n* (%)	4 (16)	5 (20)	
PAsys (mmHg)	39.6 ± 13.8	41.9 ± 13.2	0.573

MLHFQ, Minnesota living with heart failure questionnaire; GFR, glomerular filtration rate; LVEF, left ventricular ejection fraction; PAsys, systolic pulmonary arterial pressure. * Six-minute walk test could only be performed by 43 of the 50 study patients.

**Table 3 jcm-12-04458-t003:** Immunological parameters of study patients at baseline.

	Empagliflozin (*n* = 25)	No Empagliflozin (*n* = 25)	*p* Value
CRP (mg/dL)	1.1 ± 0.38	1.26 ± 0.29	0.739
IL-1β (pg/mL)	30.4 ± 8	20 ± 6.4	0.466
IFN-α2 (pg/mL)	13.1 ± 4	11.1 ± 3	0.682
IFN-γ (pg/mL)	21.3 ± 6.9	14.4 ± 5.16	0.600
TNF-α (pg/mL)	49.2 ± 16.2	39.4 ± 11.9	0.737
MCP-1 (pg/mL)	562 ± 80	598 ± 53	0.580
IL-6 (pg/mL)	21.7 ± 4.32	20.1± 2.91	0.823
IL-8 (pg/mL)	210 ± 42	174 ± 46.9	0.378
IL-10 (pg/mL)	22.8 ± 8.3	71.7 ± 52.8	0.351
IL-12p70 (pg/mL)	17.2 ± 6.7	11.6 ± 4.4	0.904
IL-17A (pg/mL)	2.11 ± 0.85	1.02 ± 0.82	0.085
IL-18 (pg/mL)	34 ± 23	119 ± 72.2	0.178
IL-23 (pg/mL)	24.2 ± 9.9	23.3 ± 8.4	0.752
IL-33 (pg/mL)	127 ± 45	89 ± 32	0.633

CRP, C-reactive protein; IL, Interleukin; IFN, Interferon; TNF, tumour necrosis factor; MCP, membrane cofactor protein. Data are presented as mean ± standard error of mean.

**Table 4 jcm-12-04458-t004:** Changes in study parameters 3 months after baseline.

	Empagliflozin (*n* = 25)			No Empagliflozin (*n* = 25)		
	Baseline	3 Month Follow-Up	*p* Value	Baseline	3 Month Follow-Up	*p* Value
MLHFQ (points)	44.2 ± 20.2	24 ± 17.7	<0.001 †	48.4 ± 23.4	41.7 ± 22	0.044
Six-minute walk test (m) *	343 ± 145	450 ± 115	<0.001 †	321 ± 157	325 ± 170	0.213
LVEF (%)	32.5 ± 8.4	40.8 ± 13.1	0.001 †	28.8 ± 8.9	35.9 ± 12.2	0.005
CRP (mg/dL)	1.1 ± 0.38	0.61 ± 0.2	0.050	1.26 ± 0.29	0.47 ± 0.10	0.022
IL-1β (pg/mL)	30.4 ± 8	27 ± 7.1	0.532	20 ± 6.4	27.1 ± 7.55	0.255
IFN-α2 (pg/mL)	13.1 ± 4	12.9 ± 3.5	0.532	11.1 ± 3	12.2 ± 2.94	0.070
IFN-γ (pg/mL)	21.3 ± 6.9	23.3 ± 6.3	0.795	14.4 ± 5.16	17.3 ± 5.5	0.093
TNF-α (pg/mL)	49.2 ± 16.2	52 ± 16.5	1.000	39.4 ± 11.9	38.9 ± 11.7	0.878
MCP-1 (pg/mL)	562 ± 80	534 ± 73	0.968	598 ± 53	614 ±64	0.696
IL-6 (pg/mL)	21.7 ± 4.32	13.7 ± 3.1	0.008 †	20.1± 2.91	19 ± 3.7	0.301
IL-8 (pg/mL)	210 ± 42	199 ± 43	0.557	174 ± 46.9	214 ± 50	0.306
IL-10 (pg/mL)	22.8 ± 8.3	20.9 ± 7.9	0.532	71.7 ± 52.8	82.9 ± 62.7	0.246
IL-12p70 (pg/mL)	17.2 ± 6.7	16 ± 7.1	0.374	11.6 ± 4.4	14 ± 4.4	0.507
IL-17A (pg/mL)	2.11 ± 0.85	1.85 ± 0.87	0.484	1.02 ± 0.82	1.89 ± 0.8	0.075
IL-18 (pg/mL)	34 ± 23	72 ± 36.6	0.144	119 ± 72.2	62 ± 29.8	0.767
IL-23 (pg/mL)	24.2 ± 9.9	20.3 ± 9.3	0.401	23.3 ± 8.4	26.4 ± 8.7	0.386
IL-33 (pg/mL)	127 ± 45	106 ± 42	0.241	89 ± 32	108 ± 36	0.071

MLHFQ, Minnesota living with heart failure questionnaire; LVEF, left ventricular ejection fraction; IL, Interleukin; IFN, Interferon; TNF, tumour necrosis factor; MCP, membrane cofactor protein. * Six-minute walk test could only be performed in 43 of the 50 study patients. † Significant (*p* < 0.05) after adjustment for multiple testing using Benjamini–Hochberg procedure. Data are presented as mean ± standard error of mean.

## Data Availability

The data presented in this study are available on request from the corresponding author. The data are not publicly available due to privacy and ethical restrictions.

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
