# Peer review of "Empagliflozin Reduces Interleukin-6 Levels in Patients with Heart Failure"

_jcm, 2023, doi:10.3390/jcm12134458_

Round 1

Reviewer 1 Report

  Groups of study  are  described as  "  Half of patients "  - it is not applicable : patients number  at each  group  should be mentioned !

 The dose of Empa is not mentioned   Methods . Only in  line 150 in  Results  is indicated in  brackets. 

If all patients  were  with diabetes ,  the explanation of 10 mg  dose application is needed. How mant patients have been withdrawn from Empa , not mentioned ?  dose  downtitration ?

 Table 3   shows the mean   concentrations of cytokines   at each group , meanwhile the same  column is  shown in Table 4    with  follow up  changes : no sence  to have it in two places .  p values  are important to  compare    with regard to  follow up changes.

 In lines 143-145  are mentioned   number of patients on  on beta blockers , ACE  inhibitors  not  in  each group , but total number   from all patients- no any importance  for this  . Comparable data are presented  already in Table 1. 

 Despite the  fact of study of  effects of Empa on many   cytokines , results are not interpreted  properly  both  at part of Results and in Discussion  of e ach of  these cytokines.

Sufficient English, no major  objectives

Author Response

Reviewer 1 - Thank you for your comments

Comments and Suggestions for Authors

  • Groups of study  are  described as  "  Half of patients "  - it is not applicable : patients number  at each  group  should be mentioned !

Answer: We agree with you. We changed the text according to your suggestion.

We included 50 inpatients with HFrEF and diabetes mellitus type 2. 25 patients received a therapy with the SGLT-2-inhibitor empagliflozin in addition to standard medication, the other 25 patients did not receive empagliflozin and were considered as control group.

  • The dose of Empa is not mentioned   Methods . Only in  line 150 in  Results  is indicated in  brackets. 
  • If all patients  were  with diabetes ,  the explanation of 10 mg  dose application is needed. How mant patients have been withdrawn from Empa , not mentioned ?  dose  downtitration ?

Answer: For type 2 diabetes, an initial dose of 10 mg empagliflozin once daily is recommended for monotherapy or combination therapy. Our patients were all patients who had never received an SGLT2 inhibitor before. We therefore chose the recommended initial dose.

We added the following statement in the methods (line 122 ):

Since all patients in this study had never received SGLT2 inhibitors before, the recommended initial dose of 10 mg empagliflozin per day was prescribed for all patients.

  • Table 3   shows the mean   concentrations of cytokines   at each group , meanwhile the same  column is  shown in Table 4    with  follow up  changes : no sence  to have it in two places .  p values  are important to  compare    with regard to  follow up changes.

Answer: We agree that the presentation of the data in Tables 3 and 4 may at first look somewhat confusing. Table 3 illustrates the concentration of cytokines at baseline and shows that there is no difference in cytokines at study inclusion in either group.

To illustrate the changes in the study parameters LVEF, quality of life, walking distance, and cytokines at baseline and after 3 months, we had to relist the baseline values. This double presentation is necessary also to perform the adjustment for multiple testing using Benjamini-Hochberg procedure.

  • In lines 143-145  are mentioned   number of patients on  on beta blockers , ACE  inhibitors  not  in  each group , but total number   from all patients- no any importance  for this  . Comparable data are presented  already in Table 1. 

Answer: We agree with you. We deleted the corresponding paragraph in the text (line 145)

(Table 1) Drug therapy included beta-blockers (n = 44), angiotensin-converting en-zymes, angiotensin II receptor blockers, or sacubitril/valsartan (n = 46), aldosterone antagonists (n = 16), insulin (n = 12), and metformin (n = 23).

  • Despite the  fact of study of  effects of Empa on many cytokines , results are not interpreted  properly  both  at part of Results and in Discussion  of each of  these cytokines.

Answer: In this study, a total of 14 immunological factors were investigated. after adjustment for multiple testing using Benjamini-Hochberg procedure, however, only IL-6 showed a significant reduction after 3 months in the group of patients receiving empaglifozin therapy.  We elaborated on this finding in the Discussion:

“As a limitation, it should be noted, that of the 14 factors analyzed, only C-reactive protein and soluble interleukin-6 showed a significant change. Interestingly, the level of C-reactive protein decreased both in the group of patients with and without empagliflozin. However, the change in C-reactive protein was no longer significant in both groups after adjustment for multiple testing (Table 4). The fact that other immunological parameters did not exhibit a change could be due to the relatively small number of patients and the relatively short duration of therapy with empagliflozin.

Nevertheless, our study confirms observations made in previous studies. A reduction in mRNA expression of interleukin-6 in empagliflozin-treated macrophages (18) and intracellular levels in human and murine myocardium (20) had already been described.

Interleukin-6 has a key role in innate immunity. The action of this cytokine is particularly related to host defense, regulation, proliferation, and differentiation of immune cells (22).  Interleukin-6, which is detectable in blood, originates from various sources, mainly mononuclear macrophages, but also T-helper cells, B-cells, vascular endothelial cells, smooth muscle cells, and fibroblasts (23).

For several years, it has been known that the expression of interleukin-6 is increased in the circulation and myocardium of patients with heart failure. Similarly, interleukin-6 is known to be associated with the progression of heart failure. Inflammation-induced remodeling of the myocardium (including an increase in apoptosis of myocytes and a decrease in contractility) is thought to be responsible (24,25). Elevated blood interleukin-6 concentrations have also been associated with increased heart failure-related mortality (26).”

We add the following statement:

It is therefore conceivable that with a larger study cohort and a longer study period, an additional significant change in further immunological parameters might be detectable.

Reviewer 2 Report

The current manuscript titled: "Empagliflozin reduces interleukin-6 levels in patients with heart failure" represents an important analysis of evolving field of Cardiology.

In my opinion, these are the adjustments which should be made to increase the value of your manuscript:

1.      Lines 21-22: The Authors indicated that SGLT2i has been shown to be beneficial in the treatment of diabetic and non-diabetic patients with heart failure with reduced ejection fraction. However, it has long been known that SGLT2i are prescribed for both patients with reduced ejection fraction and normal (e.g., EMPEROR-Preserved study, etc.). It is recommended to correct this information and add more bibliographic sources.

2.      In Introduction chapter, please, add detailed information about the pathophysiological processes linking heart failure and diabetes mellitus, and describe the relationship between inflammation and SGLT2i. Also, describe in detail already known SGLT2i mechanisms of action in heart failure.

3.      Line 75: change please “Diabetes mellitus type II” totype 2 diabetes mellitus”.

4.      Line 78: change please “Diabetes mellitus type I” totype 1 diabetes mellitus”.

5.      Why patients with a GFR < 45 ml/min/1.73 m² were excluded from the study? A contraindication for Empagliflozin administration is a GFR < 30 ml/min/1.73 m².

6.      In the Discussion section, there is not enough comparative information with other studies.

7.      In Conclusions section please highlight the practical implications of your study and its relevance to clinical practice.

8.      Add future perspectives.

9.      The manuscript contains some punctuation errors, please revise the text.

Minor editing of English language required

Author Response

Reviewer 2 - Thank you for your comments

Comments and Suggestions for Authors

The current manuscript titled: "Empagliflozin reduces interleukin-6 levels in patients with heart failure" represents an important analysis of evolving field of Cardiology.

In my opinion, these are the adjustments which should be made to increase the value of your manuscript:

  1. Lines 21-22: The Authors indicated that SGLT2i has been shown to be beneficial in the treatment of diabetic and non-diabetic patients with heart failure with reduced ejection fraction. However, it has long been known that SGLT2i are prescribed for both patients with reduced ejection fraction and normal (e.g., EMPEROR-Preserved study, etc.). It is recommended to correct this information and add more bibliographic sources.

Answer: We agree with you. The background of our study was the data and approval of the drugs at the beginning of 2020. At that time, there was no approval for the treatment of patient with heart failure without diabetes and not for the treatment of HFpEF.

At your suggestion, we changed the abstract to be more general and added to the introduction accordingly:

Abstract:

Background: Inhibition of sodium-glucose co-transporter 2 (SGLT-2) has been shown to be beneficial in the treatment of diabetic and non-diabetic patients with heart failure with reduced ejection fraction (HFrEF). The underlying mechanisms are incompletely understood. The present prospective study investigates for the first time the effect of empagliflozin on various soluble markers of inflammation in patients with reduced ejection fraction (HFrEF).

Introduction: These favourable effects have been also demonstrated in non-diabetic patients with heart failure and reduced ejection fraction (HFrEF) (5-7) and in patients with heart failure with preserved ejection fraction (HFpEF) (Anker, Solomon 8, 9).

  1. In Introduction chapter, please, add detailed information about the pathophysiological processes linking heart failureand diabetes mellitus, and describe the relationship between inflammation and SGLT2i. Also, describe in detail already known SGLT2i mechanisms of action in heart failure.

Answer: We add the following statements in the Introduction:

“Diabetes mellitus can directly damage the myocardium by causing macro- and microangiopathy of the coronary vessels. In addition to vasculopathy, several pathophysiological mechanisms of diabetic cardiomyopathy are discussed: formation of advanced glycation end products, increased oxidative stress, increased neurohumoral activation, and in particular inflammation (Park). Inflammation plays a central role in the onset and progression of heart failure (Dyck).”

Previous studies examining liver and kidney function demonstrated that SGLT2i can reduce organ inflammation (Dyck). Although it is not clear which myocardial cells are most involved in inflammation in HF, it has been demonstrated in cultured myocytes that SGLT2i is potent in reducing transcript levels of numerous markers of inflammation (Byrne). Moreover, studies in recent years suggest that SGLT2i are suitable to reduce oxidative stress and endothelial dysfunction, and to improve cardiac metabolism and energetics (Dyck).”

  1. Line 75: change please “Diabetes mellitus type II” to“type 2 diabetes mellitus”.
  2. Line 78: change please “Diabetes mellitus type I” to“type 1 diabetes mellitus”.

Answer: We changed the text as request.

  1. Why patients with a GFR < 45 ml/min/1.73 m² were excluded from the study? A contraindication for Empagliflozinadministration is a GFR < 30 ml/min/1.73 m².

Answer: We agree with the reviewer that the thresholds are lower nowadays. However, at the time of study entry, there was a clear recommendation not to use empagliflozin GFR < 45 ml/min/1.73 m². To keep the study population homogeneous, we considered this cutoff value for all patients.

  1. In the Discussion section, there is not enough comparative information with other studies.

      Answer: We added the following section: “Rodent models suggest that SGLT2 is only expressed in the proximal tubular cells of the kidney (Ghezzi C et al., J Am Soc Nephrol. 2017 Mar;28(3):802-810). Hence, it remained elusive for a long time, whether SGLT2-inhibitors may indeed have antiinflammatory effects. If so, they are not supposed to be mediated by direct effects on immune cells but rather secondary effects following improved hemodynamics, mild ketonemia or reduction of wateroverload. Our findings are in line with recent findings of the CANVAS study program. In analogy to our findings with empagloflozin, canagliflozin reduced IL6-concentrations in  In patients with type 2 diabetes at high cardiovascular risk (Koshino A, Schechter M, Sen T, et al. Interleukin-6 and cardiovascular and kidney outcomes in patients with type 2 diabetes: new insights from CANVAS. Diabetes Care 2022;45:2644-2652). Interestingly, baseline IL-6 and its 1-year change were associated with renal and cardiovascular outcomes. Although this association is not necessarily causal, it shows that IL-6 may be used as a prognostic biomarker in this context.”

  1. In Conclusions section please highlight the practical implications of your study and its relevance to clinical practice.

Answer: We added the practical implications as requested: “These findings are of practical clinical interest, since, first they show that SGLT2-inhibition has indeed antiinflammatory properties, although the transporter has not been identified in immune cells yet. They thereby contribute another piece in the puz-zle in our understanding of the complex effects of SGLT2-inhibition. Second, in line with the recent insights from CANVAS, the study identifies Interleukin-6 as a potential biomarker in the evaluation of therapeutic measures in heart failure (Koshino et al., s. o.).”

  1. Add future perspectives.

Answer: We added future perspectives after the clinical implications: “Future SGLT2-inhibitor studies should characterize this broadly available biomarker re-garding prognosis and therapy guidance in heart failure. Moreover, it will be of interest, whether SGLT2-inhibitors reduce Interleukin-6 in non-heart failure populations as well, e. g. in chronic kidney disease.”

  1. The manuscript contains some punctuation errors, please revise the text.

Answer: We changed the text as requested.

Reviewer 3 Report

Authors have a done an excellent and innovative study. However, following things should be addressed to improve the MS:

-Authors have mentioned that "...in soluble 33 interleukin-6 level (baseline 21.7 ± 21.8 pg/ml vs. 13.7 ± 15.8 pg/ml; p=0.008"; here the standard errors are higher than the main value. Any explanation? Can't you improve the data?

-I guess similar immune effects can be exerted by other anti-inflammatory drugs. Then why have you chosen empagliflozin? How was the dose determined/optimized for the current study? If lower doses were given, could there be any changes in the outcome?

-Only IL-6 levels could show improvements. Broad spectrum anti-inflammatory drugs might have better efficacy? If so, please discuss the same in the conclusion section.

Author Response

Reviewer 3

Comments and Suggestions for Authors

Authors have a done an excellent and innovative study. However, following things should be addressed to improve the MS:

-Authors have mentioned that "...in soluble 33 interleukin-6 level (baseline 21.7 ± 21.8 pg/ml vs. 13.7 ± 15.8 pg/ml; p=0.008"; here the standard errors are higher than the main value. Any explanation? Can't you improve the data?

Answer: Due to the limited sample size standard deviations are indeed high. We modified the manuscript and now present standard error of mean instead of standard deviation. Moreover, we mention this aspects in the „Limitations” section.

-I guess similar immune effects can be exerted by other anti-inflammatory drugs. Then why have you chosen empagliflozin? How was the dose determined/optimized for the current study? If lower doses were given, could there be any changes in the outcome?

Answer: Yes, of course a reduction of IL6 can be elicited by other antiinflammatory drugs as well and there are even specific drugs targeting IL6 like tocilizumab. The focus of the present study, however, was to improve our understanding of the mechanisms underlying the beneficial effects of SGLT2-inhibition. Since SGLT2 has not been identified in immune cells so far, it was controversial, whether empagliflozin has indeed an antiinflammatory effect.

-Only IL-6 levels could show improvements. Broad spectrum anti-inflammatory drugs might have better efficacy? If so, please discuss the same in the conclusion section.

Answer: We mention CANTOS and LODOCO2 as two trials showing cardiovascular benefits by anti-inflammatory drugs in the revised version of the manuscript and add the following statement:

“In recent years, there have been prospective randomized trials demonstrating that anti-inflammatory therapy leads to improved cardiovascular outcomes in patients with coronary artery disease (19,20). This may indicate a general mechanism and could improve the treatment of cardiovascular disease.”

Round 2

Reviewer 2 Report

I agree with the changes made, which significantly improve the quality of the manuscript.

Minor editing of English language required